# Factors contributing to immunization coverage among children less than 5 years in Nadowli-Kaleo District of Upper West Region, Ghana

Alice Kuuyi[1,2], Robert Kogi [2,3]*

1 Department of Disease Control, Nadowli-Kaleo District Health Directorate, Ghana Health Service, Nadowli, Ghana, 2 Department of Health Information, Asunafo South District Health Directorate, Ghana Health Service, Kukuom, Ghana, 3 Department of Social and Behavioural Sciences, School of Public Health, University of Ghana, Legon, Accra, Ghana

* robertkogi87@gmail.com

**Data Availability Statement:** All relevant data are included in the article and its supplementary information files.

## Abstract

Immunization prevents deaths from diseases such as diphtheria, tetanus, whooping cough and measles in about 2.5 million children each year worldwide. Failure to vaccinate children in the required timeframe could result in disease outbreaks among them and increase costs of living among the populations directly affected. Even though Ghana has set 95% as the target for immunization coverage, the Nadowli-Kaleo district has been below this national target. This study was conducted to identify the factors influencing low immunization coverage among children under five years in the Nadowli-Kaleo district, Ghana. An analytical cross-sectional study design was adopted for this study. Systematic sampling method was used to recruit the respondents. The data was collected using KoboCollect online data collection tool and analyzed using Stata Version 17.0. Chi-square test was used to establish the association between child's immunization status and the independent variables. Logistic regression was used to determine the degree of association. A p-value less than 0.05 was considered statistically significant. Fully immunized status among children under-five was a little above average (55.4%). Mother's or caregiver's age, marital status, occupation, and current child's birth order were significantly associated with children full immunization. Also, number of ANC visits, delivery location, and distance to health facility were significantly associated with children full immunization. A child delivered at the health facility had higher chance of 2.12 times to be fully immunized than giving birth at home [AOR = 2.12, 95%CI = 1.14–3.94, and p-value = 0.017]. The health system related factors which were statistically associated with child full immunization included time spent during immunization service and being informed when to come for the next vaccination visit. Coverage for children with fully immunized status was not very encouraging at our study site. We recommend expansion of access to maternal and child health services, encouraging institutional childbirth, and timely regular antenatal visits.

**Funding:** The authors received no specific funding for this work.

**Competing interests:** The authors have declared that no competing interests exist.

## Introduction

Immunization is a measure used in a person to make him or her resistant against an infectious disease by typically administrating a vaccine or previous infection to the wild microorganism [1]. Vaccine is given to protect the person's body system against any future infection or disease [2]. Immunization coverage, on the other hand, is referred to the percentage of individuals in a target population who are vaccinated [3]. Over the years, immunization has been proven to be a very cost-effective and life-saving intervention which protects people from needless suffering like sickness, disability, and death. Most importantly, the benefits of immunization are to all people, which target the improvement in health and life expectancy, with impact at the social, economic, global, national, and community levels [4]. There is a great contribution to be made by the increasingly interdependent communities working together to tackle diseases, which are of public health concern and can be prevented with vaccine [5].

Basic immunizations are estimated to avert 2.5 million annual child deaths globally from diphtheria, tetanus, pertussis and measles [6]. Currently, more children are getting vaccinated at the appropriate time, but approximately 20 million individuals across the globe are still not receiving vaccinations, leaving them vulnerable to severe illnesses, fatalities, handicaps, and poor health. African regions alone accounted for 8.5% of unvaccinated children in the world and almost much as in other regions combined [7]. Globally, Africa has the lowest coverage of children fully vaccinated [8]. A weakness of this system is the high dropout rate between doses. Therefore, there is a need to strengthen health care and routine immunization systems throughout the region [9].

Studies have indicated that factors that affect immunization coverage could be client-related, service-related, or health management-related. For instance, it was reported that lack of information or motivation, place of immunization not convenient for mothers, inconvenient immunization time, vaccinators absent, family problems, including maternal overwork, maternal and child illness, were identified as factors which influenced immunization coverage [10]. The level of compliance of mothers with childhood immunization is also influenced by their knowledge of childhood immunization, higher level of maternal education, birth within proximity to a medical facility and in a medical facility [11].

The need to raise child survival is a key global development goal, which is emphasized once again in the Sustainable Development Goals (2016–2030). Massive investments have been made because of these agreements to enhance the acceptability, affordability, and accessibility of child health intervention programs, particularly those related to nutrition and immunization services [12].

Enhancing the extent of immunization is crucial in advancing the health of children and diminishing the incidence of ailments and fatalities during childhood. However, immunization coverage in developing countries has been reported to be low [6]. For instance, in Ethiopia, a study conducted using a cross-sectional study design revealed that 77.4% of children received full immunization, 15.5% children were partially immunized, while some 7.1% did not receive any antigen [6]. In Mukurweni and Tetu Sub-counties, Kenya, moderate and high immunization coverages were reported [13].

Moreover, it has been found that substantial socioeconomic inequities, such as residency, wealth, educational status, and the number of children in a household, affect the coverage of immunization [14]. For instance, in Kenya, a population-based cross-sectional study conducted identified that the factors which influenced low immunization coverage included family size, location of child's birth, the level of literacy, awareness of the vaccination schedule, and whether the family lives a nomadic lifestyle [5]. Moreover, Kiptoo and colleagues noted that the income of household, proximity of the nearest healthcare facility, and whether the family resided in an urban or rural area impacted immunization coverage [5].

In addition, another study conducted in the slums of Mumbai, India, using a community-based cross-sectional approach found that inadequate time, insufficient knowledge, concerns about negative side effects, and potential income loss were among the significant factors that hindered children from receiving immunizations [15]. Moreover, a study conducted in Burkina Faso revealed that factors such as location of residence, year of visit, ethnic group, and maternal education were strongly linked to the vaccination status of children [16].

In Ethiopia, it was also noted that the distance from health facility, delivery place, and follow-up during ANC, were significance with a child being fully immunized [17]. Similarly, it was revealed in a qualitative study conducted in Ethiopia that the most shared factors among the discussants which affected routine childhood immunization service uptake were inaccessible health facilities, unfavourable attitudes of health staff, poor performance and mistreatment of healthcare personnel, inconvenient vaccination schedules, insufficient information during vaccination days, and extended waiting periods [18].

It was established in Ghana in 2014 that, nationwide, the proportion of children aged 12–23 months who did not receive all basic immunizations decreased from 79% to 77% (9). During the same period, the proportion of children without childhood vaccination increased nationally from 1% to 2% [9]. Additionally, factors related to the mother such as her occupation, educational background, age, and familiarity with vaccine-preventable illnesses and vaccination, were identified as influencers of immunization coverage in the Assin North Municipality in the Central Region of Ghana [19]. Also, it was revealed in another study conducted in the Tamale Metropolis that attaining a higher education, being a mother, having an older age, earning a higher income, having a greater number of children, living in rural areas, possessing extensive knowledge on immunization, and exhibiting a positive attitude towards immunization, all resulted in a decline in the failure to comply with all the planned immunizations [20].

The Ghana Health Service (GHS) has enhanced its Expanded Programme on Immunization (EPI) to align with the global objectives established at the 1988 World Health Assembly. In addition to vaccination coverage, the GHS now emphasizes eradicating other illnesses like poliomyelitis, while also striving to eliminate measles and neonatal tetanus. Disease surveillance and control measures have also been intensified [21]. Immunization performance has become a key health performance indicator in Ghana for the entire health sector and is monitored at all levels [21]. However, reports show that some districts record low immunization, stifling the achievement of set targets.

The national target for all antigens in Ghana is 95% coverage [22]. Penta 3 has been used as a reference indicator for measuring the coverage, quality, and utilization of childhood immunization. For the past 10 years, Ghana is able to maintain high immunization coverage rates between roughly 90–95%. However, there are reports of low-performing areas in some districts which are not achieving the 95% target with 3 doses of pentavalent vaccine [23]. This has been demonstrated in the Nadowli-Kaleo district where the immunization coverage has been consistently below the national target of 95% coverage for all antigens over the past five years. The Penta 3 coverage for Nadowli-Kaleo District from 2018 to 2021 was 71.3%, 70%, 70%, 82.0% and 86.3% respectively [24]. This poses a threat to vaccine-preventable disease outbreaks and other implications of low coverage, hence, making this study necessary.

The objective of this study was to identify the factors associated with low immunization status among children under five years in the Nadowli-Kaleo District.

## Methods

### Study design

This study adopted a cross-sectional study involving 336 mothers whose children were under five years of age and living in the Nadowli-Kaleo District. This design was chosen as the

researchers intended to gain knowledge and information on low immunization coverage within a short period in the study area.

## Study site description

The Nadowli-Kaleo District is positioned in the central part of Upper West region, with latitude between 10' 20' and 11' 30' north and longitude 3'10' and 2'10' west. It shares borders to the south with the Wa Municipal, to the west with Burkina Faso, north with Jirapa District, and to the east with the Daffiama-Bussie-Issa District. The district spans an area of 2,742.50 km$^2$, stretching from the Billi Bridge to the Dapuori Bridge, nearly 12 km away from Jirapa. The district extends from the Black Volta (Charikpong) to Daffiama from west to east. According to the 2021 Population and Housing Census, the district had a population of 77,057 with 36,993 males and 40,064 females [25]. More than half (53.35%) of the population of the district is the female group with a Total Fertility Rate (TFR) of 3.2 [26]. Finally, the Nadowli-Kaleo district has one district hospital serving as a referral centre situated in Nadowli, 10 health centres, and 145 outreach points which offer immunization services.

## Study population

All mothers or caregivers with children below the age of five years and living within the Nadowli-Kaleo District were considered in the study.

## Inclusion and exclusion criteria

The mothers or caregivers who were included in this study were those with children up to five years of age, who were residents of the district. Mothers whose children fit the inclusion criteria but were severely sick and needed medical attention were excluded.

## Sample size determination

We calculated the sample size for this study using the Cochran formula [27], n = $\frac{Z^2 pq}{d^2}$, Where n = Sample size, Z = $Z_{score}$, p = proportion estimated for immunization coverage present in the population. Coverage of 72.7% obtained in the year 2019 for Penta 3 [14] in the Nadowli-Kaleo District was used. q = 1-p, d = Level of statistical significance of 95% = (0.05). We used the assumption of a margin of error of 0.05 and a 95% confidence level.

$$\text{Sample size (n)} = \frac{(1.96)^2 \times 0.727 \times (1 - 0.727)}{(0.05)^2} = \text{n} \approx 305$$

Adding a non-respond rate of 10%, n = (305*0.10) +305 = 336. Therefore, 336 mothers with children less than five years were considered in the study.

## Sampling method

All eight sub-districts in the Nadowli-Kaleo District were considered in conducting this study. This study was conducted in health centres because they recorded a high intake of child welfare clinic (CWC) attendance in the district. The data was collected in all 8 subdistricts in the district, and all 9 health centres where CWC was conducted were purposively used.

The proportionate sample size at each CWC centre was obtained using the 2019 immunization coverage for Penta 3 per subdistrict as shown in Table 1. These proportional samples were calculated and estimated in the table.

**Table 1. Proportional allocation of study population by sub-district.**

| Sub-district | Immunization Coverage | Sub-district sample size $= \frac{\text{Sub−district coverage}}{\text{Dsitrict coverage}} \times$ study sample size |
|---|---|---|
| Charikpong | 210 | $\frac{210}{2166} \times 336 = 33$ |
| Dapour | 186 | $\frac{186}{2166} \times 336 = 29$ |
| Jang | 406 | $\frac{406}{2166} \times 336 = 63$ |
| Kaleo | 415 | $\frac{415}{2166} \times 336 = 64$ |
| Nadowli | 409 | $\frac{409}{2166} \times 336 = 63$ |
| Nanvil | 120 | $\frac{120}{2166} \times 336 = 19$ |
| Sombo | 187 | $\frac{187}{2166} \times 336 = 29$ |
| Takpo | 233 | $\frac{233}{2166} \times 336 = 36$ |
| **Total** | **2166** | **336** |

To obtain the appropriate sample size for this study, a systematic sampling technique was used. Systematic sampling is a type of random sampling where the initial unit is chosen randomly using random numbers, and subsequent units are chosen automatically based on a pre-determined pattern [28]. Initially, the researchers collaborated with the heads of different healthcare facilities to determine the count of children registered in the Child Welfare Clinic (CWC) registers who fell within the age range of 0 to 59 months. The registers were numbered serially to form the sampling frame, then the proportionate sample to be obtained was used to divide it. Secondly, the total number of children in each register was divided by the respective samples for each of the subdistricts to calculate the interval that was used when selecting the children. Numbers on the ballot papers were given from 1 to the interval number that was obtained. That number corresponded to the mother who had the number that comes with the facility. The sampling interval number was then added to obtain every nth number.

## Data collection procedure

Four people were trained for two days from 24th-25th of April 2021, to help collect the data. The training was done using both English and Dagaare; demonstration was done using Dagaare which aided in effective data collection. The essence of the research and procedure were explained to them. They were encouraged to ask questions to clarify any doubts.

Interviewer-administered pretested questionnaire was administered to respondents using KoboCollect online data collection tool. The questionnaire was made up of five sections: socio-demographic characteristics of respondents, knowledge-based questions of caregivers/mothers on child immunization, and attitude of caregivers/mothers towards child immunization. To obtain health system related factors, mothers were asked questions such as long waiting period, attitude of health workers towards them, special education to them on immunization when they came, information on side effects, how to take care of it, as well as distance to the nearest facility. Data was collected on the days' caregivers/mothers attended the child welfare clinic. The methods employed to distribute the questionnaires involved individualized interaction, where the questions were asked directly and answered by the respondents. On average, it took approximately 15 minutes for each respondent to complete the questionnaire.

## Data analysis

The data obtained was processed and imported from Excel sheet into the STATA 17.0 software for analysis. The focus of the study was to determine the immunization status of children

under the age of 5 years, which was classified as either fully immunized or partially immunized. In addition, the researchers analyzed independent variables, including factors related to the mother's individual and community characteristics such as age, education, occupation, antenatal care visits, and the birth order of the child as presented in S1 Data. Categorical variables were presented as proportions and frequencies. The dependent/outcome variable was a dichotomous variable (Fully or partially immunized). Immunization was defined as follows: 1 is full immunization if the child received both doses of the vaccine as recommended and partially as 0 if a child did not receive any dose or received one dose of vaccine before the study. The association between immunization status and independent variables was analyzed using the chi-square test. Additionally, logistic regression was employed to assess the factors influencing children full immunization. To determine significant associations, a confidence interval of 95% and a p-value below 0.05 were used as criteria.

## Ethical issues

Ethical clearance was sought from the University of Health and Allied Sciences (UHAS) Ethics Review Committee with approval number **UHAS-REC A.9 [126] 20–21**. Permission was sought from the Upper West Regional Health Directorate and the District Health Directorate of the Nadowli-Kaleo District. Formal verbal consent was obtained from all the participants involved in the study. The verbal consent process was documented and witnessed in accordance with the Ethics Committee's approved guidelines. We developed a verbal consent script detailing the study's purpose, procedures, risks, benefits, and participant rights. During the consent process, researchers explained these details to participants, allowing time for questions and clarifications. Researchers also verified participants' understanding by having them summarize key study details. The entire procedure, including the script was approved by the Ethics Committee, ensuring adherence to ethical standards and protection of participants right throughout the study.

The recruited participants were given complete information regarding the research's aim, and their privacy as well as confidentiality were guaranteed. It is worth noting that no mother or caretaker was compensated for their involvement in the study. However, they were thanked for their acceptance to participate in the study.

Any interaction during data collection was done while wearing a facemask and maintaining social distancing as ways of observing the Covid-19 protocol. Research assistants and participants also used alcohol-based hand sanitizers throughout the period of the data collection.

**Timeliness.** This project started on 24[th] April 2021 and ended on 20[th] November 2021.

## Results, discussion, conclusions

### Results

The results in Table 2 showed that less than half (42.9%) of the respondents (mothers) were at most 30 years old. This was immediately followed by those aged 31–35 years (42.0%). More than half (57.4%) of the children in this study were within 24–59 months old and a little above average (51.5%) were females. Moreover, more than two-fifth (43.8%) of the children in this study were first-born.

Among the mothers, a greater proportion (73.5%) of them were married and less than two-third (64.3%) of the mothers were Christians. Moreover, approximately two-third (66%) of the respondents were Dagaabas. A little above half (51.2%) of the respondents attained primary or JHS education. The results further showed that a little above average (52.7%) of the respondents were unemployed and a vast most majority (93.8%) of them were rural dwellers. In addition, less than two-fifth (44.1%) of the respondents earned from Gh₵100.00- <Gh₵500.00

**Table 2. Socio-demographic characteristics of mothers/caregivers of children under 5 years in Nadowli-Kaleo District, Upper West Region, Ghana.**

| Age of mother/caregiver | Frequency (N = 336) | Percentage (%) |
| --- | --- | --- |
| < = 30 | 144 | 42.9 |
| 31–35 | 141 | 42.0 |
| 36–40 | 42 | 12.5 |
| 41 and above | 9 | 2.7 |
| **Age of child (Months)** | | |
| 0–11 months | 13 | 3.9 |
| 12–23 months | 130 | 38.7 |
| 24–59 months | 193 | 57.4 |
| **Sex of child** | | |
| Female | 173 | 51.5 |
| Male | 163 | 48.5 |
| **Birth order of child** | | |
| 1st | 147 | 43.8 |
| 2nd | 89 | 26.4 |
| ≥3rd | 100 | 29.8 |
| **Marital status** | | |
| Married | 247 | 73.5 |
| Single | 89 | 26.5 |
| **Religion** | | |
| Christian | 216 | 64.3 |
| Muslim | 120 | 35.7 |
| **Ethnicity** | | |
| Akan | 15 | 4.5 |
| Dagaao | 223 | 66.4 |
| Sissala | 26 | 7.7 |
| Waale | 72 | 21.4 |
| **Level of Educational** | | |
| None | 65 | 19.4 |
| Primary/JHS | 172 | 51.2 |
| SHS | 43 | 12.8 |
| Tertiary | 56 | 16.7 |
| **Occupation** | | |
| Government employee | 59 | 17.6 |
| Self employed | 100 | 29.8 |
| Unemployed | 177 | 52.7 |
| **Residential status** | | |
| Rural | 315 | 93.8 |
| Urban | 21 | 6.3 |
| **Monthly income** | | |
| <100 | 122 | 36.3 |
| 100- <500 | 148 | 44.1 |
| 500–1000 | 31 | 9.2 |
| >1000 | 35 | 10.4 |
| **Number of ANC visits** | | |
| No ANC visit | 18 | 5.3 |
| <4 times | 181 | 53.9 |

(*Continued*)

**Table 2.** (Continued)

| Age of mother/caregiver | Frequency (N = 336) | Percentage (%) |
|---|---|---|
| ≥4 times | 137 | 40.8 |
| **Place of delivery** | | |
| Home | 66 | 19.6 |
| Health facility | 270 | 80.4 |

monthly. Moreover, above average (53.9%) of the mothers in this study were found to make four ANC visits. Finally, majority (80.4%) of the mothers delivered in a health facility.

More than average (55.4%) of the respondents' children in this study were found to be fully immunized as indicated in Fig 1.

It is found in Table 3 that below average (45.7%) of the respondents whose children were fully immunized were within 31–35 years and majority (80.1%) of the women who were married had their children fully immunized. Moreover, most (62.4%) Christians had their children fully immunized. Almost two-third (65.6%) of the respondents who belonged to the Dagaaba ethnic group had their children fully immunized. Furthermore, a higher proportion (95.7%) of the respondents who were rural dwellers had their children fully immunized.

Moreover, less than half (48.4%) of the respondents who attained either primary or JHS education had their children fully immunized, and a little above half (51.1%) of the unemployed mothers were found to have their children fully immunized and majority (88.7%) of those who earned less than GH₵500.00 a month had their children also fully immunized. Moreover, majority (94.7%) of the children who were above 12 months of age were fully immunized. Additionally, a little above average (51.6%) of the children who were females were fully immunized, while less than two-fifth (39.3%) of the first-born children were fully immunized.

Furthermore, it was found in this study that the factors which were significantly associated with child's full immunization were mother's or caregiver's age, marital status, occupation, and birth order of the current child [$X^2$(P-value = 8.57(0.014), 9.31(0.002), 7.82(0.020), and 7.90(0.019)] respectively.

The results in Table 4 showed that less than half (47.4%) of the respondents who made at least one ANC attendance had their children fully immunized. For respondents who delivered in the health facility, majority (87.4%) of them had their children fully immunized. Mothers who walked at most 5 kilometres to the nearest health facility had most of their children being fully immunized (88.9%). Additionally, the results further depicted that number of ANC visits,

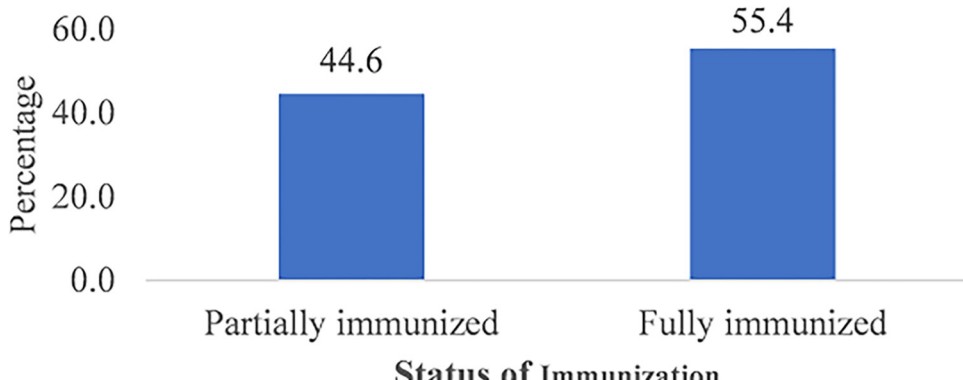

**Fig 1. Immunization status of children.**

**Table 3. Association between respondents' socio-demographic characteristics and child immunization status.**

| Variables | Immunization status | | Pearson Chi-square | p-value |
|---|---|---|---|---|
| | Partially immunized (n = 146) | Fully immunized (n = 190) | | |
| **Age (Mother/caregiver)** | | | | |
| ≤30 | 77 (51.3) | 67 (36.0) | 8.57 | 0.014 |
| 31–35 | 56 (37.3) | 85 (45.7) | | |
| 36 and above | 17 (11.3) | 34 (18.3) | | |
| **Marital status** | | | | |
| Married | 98 (65.3) | 149 (80.1) | 9.31 | 0.002 |
| Single | 52 (34.7) | 37 (19.9) | | |
| **Religion** | | | | |
| Christian | 100 (66.7) | 116 (62.4) | 0.67 | 0.413 |
| Muslim | 50 (33.3) | 70 (37.6) | | |
| **Ethnicity** | | | | |
| Dagaao | 101 (67.3) | 122 (65.6) | 1.31 | 0.520 |
| Sissala | 15 (10.0) | 26 (14.0) | | |
| Waale | 34 (22.7) | 38 (20.4) | | |
| **Level of Educational** | | | | |
| None | 26 (17.3) | 39 (21.0) | 6.92 | 0.075 |
| Primary/JHS | 82 (54.7) | 90 (48.4) | | |
| SHS | 24 (16.0) | 19 (10.2) | | |
| Tertiary | 18 (12.0) | 38 (20.4) | | |
| **Occupation** | | | | |
| Government employee | 17 (11.3) | 42 (22.6) | 7.82 | 0.020 |
| Self employed | 51 (34.0) | 49 (26.3) | | |
| Unemployed | 82 (54.7) | 95 (51.1) | | |
| **Residential status** | | | | |
| Rural | 137 (91.3) | 178 (95.7) | 2.70 | 0.100 |
| Urban | 13 (8.7) | 8 (4.3) | | |
| **Monthly income** | | | | |
| <500 | 140 (93.3) | 165 (88.7) | 2.12 | 0.145 |
| ≥500 | 10 (6.7) | 21 (11.3) | | |
| **Age of child (Months)** | | | | |
| ≤12 | 11 (7.5) | 10 (5.3) | 0.73 | 0.394 |
| >12 | 135 (92.5) | 180 (94.7) | | |
| **Sex of children** | | | | |
| Female | 75 (51.4) | 98 (51.6) | 0.00 | 0.970 |
| Male | 71 (48.6) | 92 (48.4) | | |
| **Birth order of current child** | | | | |
| 1st | 74 (49.3) | 73 (39.3) | 7.90 | 0.019 |
| 2nd | 43 (28.7) | 46 (24.7) | | |
| ≥3rd | 33 (22.0) | 67 (36.0) | | |

place of delivery, and distance to health facility were all factors which had statistically significance association with child full immunization [$X^2$(P-value) = 10.13(0.006), 13.62(P<0.001) and 4.38(0.036)] respectively.

The results in Table 5 also showed that a little above half (52.1%) of the respondents who indicated that they were never screamed at by health worker (s) when they were late during immunization sessions or services had their children fully immunized. Moreover, less than

**Table 4. Association between children's immunization status and community-related factors.**

| Variables | Immunization status | | Pearson Chi-square | p-value |
|---|---|---|---|---|
| | Partially immunized (n = 146) | Fully immunized (n = 190) | | |
| **Number of ANC visits** | | | | |
| No ANC visit | 11 (7.5) | 10 (5.2) | 10.13 | **0.006** |
| ≤3 times | 89 (61.0) | 90 (47.4) | | |
| 4+ times | 46 (31.5) | 90 (47.4) | | |
| **Place of delivery** | | | | |
| Home | 42 (28.8) | 24 (12.6) | 13.62 | **P<0.001** |
| Health facility | 104 (71.2) | 166 (87.4) | | |
| **Distance to health facility** | | | | |
| >5KM | 28 (19.2) | 21 (11.1) | 4.38 | **0.036** |
| ≤ 5KM | 118 (80.8) | 169 (88.9) | | |

two-third (60.5%) of the mothers who spent less than 30 minutes in immunization centres or sessions had their children fully immunized. In addition, a little above three-quarters (76.8%) of the mothers who were educated on how to manage vaccine side effects also had their children fully immunized. Lastly, the majority (80.5%) of mothers who were informed about their next CWC visit also had their children fully immunized.

In addition, the health system related factors which were found to be statistically associated with child full immunization included time spent at immunization service [$X^2$(P-value) = 8.01 (0.018)] and being informed when to come for next vaccination [$X^2$(P-value) = 5.13(0.024)].

When multivariate analysis was performed as shown in Table 6, it was noted that there was no association between all variables and child full immunization except place of delivery, which showed that children whose places of delivery was the health facility were 2.12 times more likely to be fully immunized compared to children who were born at home [AOR = 2.12; 95%CI = (1.14–3.94); p-value = 0.017].

## Discussion

Childhood immunization is crucial in stopping numerous illnesses and preventing a global estimated 2 to 3 million deaths [29]. However, despite the effectiveness of vaccines, vaccine-

**Table 5. Association between child immunization status and health system-related factors.**

| Variables | Immunization status | | Pearson Chi-square | p-value |
|---|---|---|---|---|
| | Partially immunized (n = 146) | Fully immunized (n = 190) | | |
| **Was screamed at by health worker when late for immunization** | | | | |
| Yes | 62 (42.5) | 91 (47.9) | 0.98 | 0.322 |
| No | 84 (57.5) | 99 (52.1) | | |
| **Time spent at immunization service** | | | | |
| <30 minutes | 82 (56.2) | 115 (60.5) | 8.01 | **0.018** |
| 30 minutes to 1 hour | 41 (28.1) | 63 (33.2) | | |
| > 1 hour | 23 (15.7) | 12 (6.3) | | |
| **Educated how to manage vaccines side effects** | | | | |
| Yes | 44 (30.1) | 44 (23.2) | 2.08 | 0.149 |
| No | 102 (69.9) | 146 (76.8) | | |
| **Being informed when to come for next vaccination** | | | | |
| Yes | 102 (69.9) | 153 (80.5) | 5.13 | **0.024** |
| No | 44 (30.1) | 37 (19.5 | | |

**Table 6. Factors associated with immunization status among children under five years old in Nadowli-Kaleo District.**

| | Immunization status | | Crude Odd ratio | Adjusted odd ratio |
|---|---|---|---|---|
| Variables | Partially immunized | Fully immunized | COR(95% CI)p-value | AOR(95%CI)p-value |
| **Age (Mother/caregiver)** | | | | |
| ≤30 | 77 (51.3) | 67 (36.0) | **Reference** | **Reference** |
| 31–35 | 56 (37.3) | 85 (45.7) | 1.70(1.06–2.72)**0.027** | 1.29(0.72–2.31)0.399 |
| 36 and above | 17 (11.3) | 34 (18.3) | 2.38(1.21–4.67)**0.012** | 1.58(0.66–3.79)0.304 |
| **Marital status** | | | | |
| Married | 98 (65.3) | 149 (80.1) | 2.29(1.40–3.75)**0.001** | 1.33(0.72–2.45)0.356 |
| Single | 52 (34.7) | 37 (19.9) | **Reference** | **Reference** |
| **Occupation** | | | | |
| Government employee | 17 (11.3) | 42 (22.6) | 2.17(1.14–4.13)**0.019** | 1.57(0.76–3.24)0.220 |
| Self employed | 51 (34.0) | 49 (26.3) | 0.77(0.47–1.27)0.308 | 0.77(0.45–1.32)0.342 |
| Unemployed | 82 (54.7) | 95 (51.1) | **Reference** | **Reference** |
| **Birth order of current child** | | | | |
| 1st | 74 (49.3) | 73 (39.3) | **Reference** | **Reference** |
| 2nd | 43 (28.7) | 46 (24.7) | 1.10(0.65–1.87)0.713 | 0.83(0.45–1.52)0.544 |
| ≥3rd | 33 (22.0) | 67 (36.0) | 2.20(1.29–3.74)**0.004** | 1.41(0.70–2.82)0.331 |
| **Number of ANC visits** | | | | |
| No ANC visit | 11 (7.5) | 10 (5.2) | **Reference** | **Reference** |
| ≤3 times | 89 (61.0) | 90 (47.4) | 1.62(0.60–4.38)0.338 | 1.10(0.38–3.20)0.864 |
| 4+ times | 46 (31.5) | 90 (47.4) | 3.11(1.13–8.55)**0.028** | 1.42(0.46–4.38)0.547 |
| **Place of delivery** | | | | |
| Home | 42 (28.8) | 24 (12.6) | **Reference** | **Reference** |
| Health facility | 104 (71.2) | 166 (87.4) | 2.79(1.60–4.88)**0.000** | 2.12(1.14–3.94)**0.017** |
| **Distance to health facility** | | | | |
| >5KM | 28 (19.2) | 21 (11.1) | **Reference** | **Reference** |
| ≤5KM | 118 (80.8) | 169 (88.9) | 1.91(1.03–3.52)**0.039** | 1.66(0.83–3.32)0.152 |
| **Time spent at immunization service** | | | | |
| <30 minutes | 82 (56.2) | 115 (60.5) | 2.69(1.32–6.56)**0.010** | 1.93(0.85–4.41)0.119 |
| 30 minutes to 1 hour | 41 (28.1) | 63 (33.2) | 2.95(1.32–6.56)**0.008** | 2.28(0.97–5.37)0.059 |
| > 1 hour | 23 (15.7) | 12 (6.3) | **Reference** | **Reference** |
| Information on next vaccination section | | | | |
| Yes | 102 (69.9) | 153 (80.5) | 1.78(1.08–2.95)**0.024** | 1.29(0.73–2.29)0.379 |
| No | 44 (30.1) | 37 (19.5 | **Reference** | **Reference** |

preventable diseases still cause around 2.5 million childhood deaths annually, with roughly 1.5 million of those deaths occurring in developing countries among children under the age of 5 [30]. The main aim of conducting this study was to identify the factors associated with low immunization among children under five years in the Nadowli-Kaleo District.

The current study found that a little over half of respondents had fully immunized their children, a coverage lower than the Ghana national target of 95% [22]. This indicates high dropout rates for some antigens and suggests poor performance and low access to immunization services. Comparatively, a study in Ethiopia reported higher proportion of 77.4% full immunization and 15.5% partial immunization [6], whereas another Ethiopian study showed only 38.3% overall full immunization coverage [31]. This is an indication that there are still high dropout rates of some antigens and could be indicative of poor performance and low access to immunization services. A similar survey in Techiman Municipality of Ghana indicated that 89.5% of children were fully vaccinated [32]. These variations highlighted

differences in healthcare access and immunization practices across regions and the differences between the findings of this current study and the previous studies can be attributed to a combination of geographic, socioeconomic, healthcare system, policy, cultural, behavioural, and methodological factors.

This study identified significant sociodemographic factors associated with complete child immunization, such as maternal age, marital status, occupation, and birth order. According to the current study, age, marital status, occupation of the mother or caregiver, and birth order of the child showed a significant association with child full immunization in the study area. This may be because mothers aged 19 years and above may have become more aware of the significance of vaccination, leading to a decrease in the number of unvaccinated children. Additionally, the detrimental effects that result from not being immunized may have contributed to this positive trend [33]. Moreover, a likely reason regarding maternal occupation would be that mothers who might be engaged in one occupation or the other would not have to depend on their partners for transportation costs to take their children for immunization. The educational level of a mother could be improved by being married, as married mothers' access to education may be better than single mothers with different responsibilities and may prioritize their children's needs over their education [34]. If both partners collaborate to improve their child's health, the supportive role of the partner may also boost the mother's knowledge. This finding aligns with research from Nigeria and the Assin North Municipality in Ghana, which also emphasized the influence of maternal age, marital status, and occupation on immunization completion [19, 35]. However, it contrasts with a study from Burkina Faso, where immunization status was linked to factors like dwelling place, ethnic background, and maternal education [16].

This study also highlighted the role of antenatal care in child immunization. Nearly half of the participants had fully immunized their children, with a significant association between at least one ANC visit and child full immunization service. Mothers delivering in health facilities were more likely to fully immunize their children, underscoring the importance of educating mothers about immunization benefits during ANC visits. Furthermore, this implied that, mothers who attended health care facilities regularly during pregnancy may have received childhood immunization counseling, which encouraged them to accept and prioritize the importance of timely childhood immunization. This supported the finding from Kenya where a child born in a health facility was five times more likely to be vaccinated than a child delivered at home [5]. Similarly, in this study, mothers living closer to health facilities were more likely to fully immunize their children, a finding consistent with research in Ethiopia linking distance to health facilities with full immunization status [6].

Behavioural factors also played a significant role in this study. Mothers not scolded by healthcare workers had higher rates of child full immunization, although this was not statistically significant. The time spent at immunization centers was statistically associated with full immunization status, with shorter waiting times leading to higher rates of full immunization. This supported the findings a study from rural Nigeria reported, where long waiting times were a barrier to full immunization [36]. This current study finding may encourage mothers to take their children for immunization services.

Education on managing vaccine side effects, although not statistically significant, was associated with higher full immunization rates. Informing mothers about their next clinic visit was significantly linked to full immunization, aligning with findings from a study in Mumbai, India, where poor awareness and fear of adverse events were major barriers to immunization services [15].

In summary, this study found that more than half of children in the Nadowli-Kaleo District were fully immunized. Key demographic factors associated with full immunization included

the age of the mother or caregiver, marital status, and religious and ethnic backgrounds, with higher immunization rates among mothers aged 31–35 years, married, Christians, and the Dagaaba ethnic group. Rural dwellers, those with primary or junior high school education, unemployed mothers, and low-income earners per month also showed full immunization service. Additionally, children over 12 months old, females, and first-borns were more likely to be fully immunized. Significant healthcare-related factors included at least one ANC visit, delivery in a healthcare facility, and proximity to health facilities. Furthermore, positive interactions with healthcare workers, shorter waiting times, education on managing vaccine side effects, and being informed about the next vaccination visit were associated with full immunization service. Multivariate analysis revealed that children born in healthcare facilities were 2.12 times more likely to be fully immunized than those born at home.

In conclusion, the immunization status of fully immunized children in the Nadowli-Kaleo District was below the Ghana national target of at least 95%. The study demonstrated significant mixed disparities which influenced children full immunization including demographic, socioeconomic, and healthcare access factors. This study also found that the factors which were significantly associated with child full immunization were age of mother or caregiver, marital status, occupation, and birth order of the current child. Children who were delivered in a healthcare facility had about twice chance of receiving full immunization as compared to those who were born at homes. The study also identified health system-related factors that had a significant association with full immunization of children including time spent at the immunization sessions and being informed when to come for the next vaccination.

We therefore recommend that the Ghana Health Service through the Nadowli-Kaleo District Health Directorate should intensify efforts to enhance access to healthcare facilities, and promote institutional deliveries, as children born in healthcare facilities are significantly more likely to receive full immunization. In addition, the District Health Directorate should strengthen ANC services by encouraging ANC visits and ensuring mothers are well-informed about immunization schedules.

## Supporting information

**S1 Data.**
(XLS)

## Acknowledgments

Many thanks go to the District Director of Health Service, Nadowli-Kaleo District, the In-charges of all facilities who assisted us in obtaining the data, and all our study participants (mothers/caregivers), for their cooperation and willingness to take part in the study, as well as the data collection assistants, for helping us administer the questionnaires successfully.

## Author Contributions

**Conceptualization:** Alice Kuuyi.

**Data curation:** Alice Kuuyi, Robert Kogi.

**Formal analysis:** Alice Kuuyi, Robert Kogi.

**Methodology:** Alice Kuuyi, Robert Kogi.

**Resources:** Alice Kuuyi.

**Validation:** Alice Kuuyi, Robert Kogi.

**Writing – original draft:** Alice Kuuyi, Robert Kogi.

**Writing – review & editing:** Alice Kuuyi, Robert Kogi.

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
