## [Decision Letter · Decision Letter 0]

29 Apr 2024

PGPH-D-24-00090

Factors contributing to immunization coverage among children less than 5 years in Nadowli-Kaleo District of Upper West Region, Ghana

Dear Dr. Kogi,

Thank you for submitting your manuscript to PLOS Global Public Health. After careful consideration, we feel that it has merit but does not fully meet PLOS Global Public Health’s publication criteria as it currently stands. Therefore, we invite you to submit a revised version of the manuscript that addresses the points raised during the review process.

The manuscript has been evaluated by two reviewers, and their comments are available below. Both reviewers indicate that work should be done to improve the reporting of this study. Specifically, revisions to improve the presentation of the study objectives and conclusions as well as suggestions to improve the statistical reporting.

We look forward to receiving your revised manuscript.

Kind regards,

Emma Campbell, Ph.D

Staff Editor

Journal Requirements:

1. In the ethics statement in the Methods, you have specified that verbal consent was obtained. Please provide additional details regarding how this consent was documented and witnessed, and state whether this was approved by the IRB.

Additional Editor Comments (if provided):

Reviewers' comments:

Reviewer's Responses to Questions

**Comments to the Author**

1. Does this manuscript meet PLOS Global Public Health’s publication criteria? Is the manuscript technically sound, and do the data support the conclusions? The manuscript must describe methodologically and ethically rigorous research with conclusions that are appropriately drawn based on the data presented.

Reviewer #1: Yes

Reviewer #2: Partly

2. Has the statistical analysis been performed appropriately and rigorously?

Reviewer #1: Yes

Reviewer #2: Yes

3. Have the authors made all data underlying the findings in their manuscript fully available (please refer to the Data Availability Statement at the start of the manuscript PDF file)?

Reviewer #1: Yes

Reviewer #2: Yes

4. Is the manuscript presented in an intelligible fashion and written in standard English?

Reviewer #1: Yes

Reviewer #2: Yes

5. Review Comments to the Author

Reviewer #1: Review of Article - Manuscript Number PGPH-D-24-00090

PLOS ONE, February 2024

Article title: Factors contributing to immunization coverage among children less than 5 years in Nadowli-Kaleo District of Upper West Region, Ghana

The authors sought to identify the factors that are associated with low childhood immunization coverage among mothers or caregivers of children under five years of age. The findings are important for public health and has implications for control of vaccine preventable diseases.

The manuscript has described methodologically and ethically a rigorous research with conclusions that are based on the data presented.

There are however a few minor errors in the paper that must be corrected to make it publishable.

Find below my specific comments,

Title and abstract

The title of this manuscript is appropriate and concise.

Lines 20-21: – Here, the authors should qualify the immunization status e.g., fully immunized status

Line 29 – Nothing was said about immunization coverage in the abstract hence the conclusion should be on fully immunization status e.g., coverage for children with fully immunized status is not very encouraging,

Background

Lines 80-82: – Probable over the past five years would make it clearer.

Methods

Sample Size Determination

Lines 155-156: - Some minor correction needed here in the word used to begin the sentence,

Sampling method

Lines 179-180: - To remove the word ‘be’,

Ethical Issues

Lines 213-214: - Upper West Regional Directorate Should be Upper West Regional Health Directorate.

Lines 222-223: - The word ‘were’ should be removed for the sentence to read ‘Research assistants and participants also used alcohol-based hand sanitizers throughout the period of the data collection.

Results

Lines 231-232: - First born should be firstborns or written with a hyphen.

Lines 237-238: - Kindly add monthly if this is their monthly earnings.

Table 2 title is not complete – Should be Socio-demographic characteristics of mothers/caregivers of children under 5 years in Upper west Region, Ghana

Lines 246-248: - The concluding part should be rephrased to respondents whose children were fully immunized were within 31-35 years so that it does not make it looks like you are referring to immunization status of the mothers or caretakers.

Table 3 Association between respondents’ socio-demographic characteristics and child full immunization – Child full immunization should be changed to immunization status since the table is looking at both full and partial immunization status.

Lines 274-276:– late during immunization should be late during immunization sessions or services.

Lines 276-277: - Add centres or sessions to immunization – e.g. mothers who spent less than 30 minutes in immunization centres or sessions.

Table 6 – According to literature the term predictors require a more statistically robust analysis and hence the authors could change predictors to factors associated with immunization status.

Discussion

Lines 301-302: - The word ‘our’ study should be removed and immunized fully should be fully immunized. Also, I would replace majority with a little over half of respondents.

Line 302: - I would say this present study’s coverage of fully immunized children is lower, compared to the Ghana national…

Lines 305-306 – 77.4 % cannot be lower than 55.5% so kindly correct the sentence

Line 315 – Add years to 19 i-e 1years and above

Lines 345-346 – The authors are comparing a child born in a health facility to those delivered at home but the last part of the sentence doesn’t make it look like the comparison is with home deliveries.

Lines 349-352 – Kindly check the sentence again

Lines 355-357 – should be during immunization sessions or services

Lines 362-364 – Should be immunization centres

Line 376 – I would suggest to rephrase as in conclusion, the immunization status of fully immunized children….

Line 385 – Included should be including

Acknowledgement: - Authors could use passive voice or use we/us in place of I/my.

Reviewer #2: The authors tried to examine the determinants of immunization coverage in Nadowli-Kaleo District of Upper West Region of Ghana. This is is a welcome development as it adds an important finding to the body of literature in public health which is still not at the level it should be in most LMICs of the world. The conceptualization was okay and the data analyst(s) did a good job. However, the article suffered from a few main components of a scientific research of this nature that are needed before it can move forward which was the reason I actually decided that the review is major. I have about 25 comments in the attached reviewed copy but I will just concentrate on 7 or 8 major ones. The minor corrections can be looked up from the attached and the corrections effected.

1. There is no scientific writing where the main objective or objectives to be achieved would not be stated. Even though authors' background to the study and literature were clear enough. One can easily state their objectives on their behalf but that is not done. The objectives to be achieved in this article was CONSCUPIOUSLY MISSING! This is not acceptable in any scientific writing no matter how it has been implied in the background to the study. This needs to be clearly stated somewhere towards the end of the background to the study.

2. Similarly, this article ended with the discussion section. Every scientific article must have a conclusion section where findings are summarised and a conclusion is drawn. This again was missing from this article. Drawing conclusions and making recommendations within discussion section does not go well with scientific writing of this nature.

3. Many results were repeated inside the discussion even with statistical data again which also is not good. In standard discussion, you discuss the important or significant results vis a vis existing studies and not necessarily repeating your results all over again. Then, you engage existing studies in comparison with your findings not just saying that your study agree or disagree with a particular study. This is not enough.

4. Your recommendations were mere pieces of advice which did not emanate from findings from your study. Authors should know the differences between suggestions and recommendations. Your recommendations should emanate from your findings and not just stating things you feel should be done.

5. It is surprising that the authors were reporting data in the range of 40-50% of the respondents and called that "the majority" and at the same time reporting those in the region of 70% and above still as the majority. Please look into this. Data that are less than 50% of the total population of the respondents should never be reported as the majority. For instance, data in the region of 50-55% of the total population could be called " more than half", "above average" and so on while you report those that are more than 80% of the total population as the majority. Authors should seriously look into this.

6. In modern statistical reporting and as a rule of thumb, it is not okay to report both absolute numbers and percentages in the same study. You either concentrate on reporting absolute numbers or percentages. In most cases, reporting percentages is better in scientific articles of this nature. This is also APA 6th and 7th editions standard.

7. In the last statement of the abstract, authors stated that they "suggest". In scientific writing, you don't suggest. You kindly recommend based on the outcome of your results, not just mere suggestions for the sake of it

8. Writing a good background to the study requires some technicality. it is better if the authors highlight all the evidences or factors affecting immunization uptake or coverage from developing countries before finally ending the background to the study with the situation in Ghana since that is their main focus. It is not good enough if you have spoken about Ghana before ending your background to the study with reports from another country before stating your objectives

-Other comments are well documented in the attached article around the areas which are yellow-inked.

6. PLOS authors have the option to publish the peer review history of their article (what does this mean?). If published, this will include your full peer review and any attached files.

**Do you want your identity to be public for this peer review?** For information about this choice, including consent withdrawal, please see our Privacy Policy.

Reviewer #1: **Yes: **Mavis Pearl Kwabla

Reviewer #2: No

---

## [Decision Letter · Decision Letter 1]

15 Jul 2024

Factors contributing to immunization coverage among children less than 5 years in Nadowli-Kaleo District of Upper West Region, Ghana

PGPH-D-24-00090R1

Dear Mr Kogi,

We are pleased to inform you that your manuscript 'Factors contributing to immunization coverage among children less than 5 years in Nadowli-Kaleo District of Upper West Region, Ghana' has been provisionally accepted for publication in PLOS Global Public Health.

You'll note below there is a final suggested revision from one of the reviewers; you are free to incorporate this suggestion during this final edit stage, but it's not required. 

Best regards,

Julia Robinson

Executive Editor

Reviewer Comments (if any, and for reference):

Reviewer's Responses to Questions

**Comments to the Author**

1. If the authors have adequately addressed your comments raised in a previous round of review and you feel that this manuscript is now acceptable for publication, you may indicate that here to bypass the “Comments to the Author” section, enter your conflict of interest statement in the “Confidential to Editor” section, and submit your "Accept" recommendation.

Reviewer #1: All comments have been addressed

Reviewer #2: (No Response)

2. Does this manuscript meet PLOS Global Public Health’s publication criteria? Is the manuscript technically sound, and do the data support the conclusions? The manuscript must describe methodologically and ethically rigorous research with conclusions that are appropriately drawn based on the data presented.

Reviewer #1: Yes

Reviewer #2: Yes

3. Has the statistical analysis been performed appropriately and rigorously?

Reviewer #1: Yes

Reviewer #2: Yes

4. Have the authors made all data underlying the findings in their manuscript fully available (please refer to the Data Availability Statement at the start of the manuscript PDF file)?

Reviewer #1: Yes

Reviewer #2: Yes

5. Is the manuscript presented in an intelligible fashion and written in standard English?

Reviewer #1: Yes

Reviewer #2: Yes

6. Review Comments to the Author

Reviewer #1: All comments raised in the initial review have been adequately addressed by the authors.

Reviewer #2: The authors have addressed all of the comments I raised in the previous version of the manuscript except one. The authors still statistically referred to 53.3% on page 8 of the current manuscript as the majority. 53.3 can be referred to as more than half but not as the majority. Authors should double-check this.

Every other comment I made has been attended to.

7. PLOS authors have the option to publish the peer review history of their article (what does this mean?). If published, this will include your full peer review and any attached files.

**Do you want your identity to be public for this peer review?** For information about this choice, including consent withdrawal, please see our Privacy Policy.

Reviewer #1: **Yes: **Dr. Mavis Pearl Kwabla

Reviewer #2: **Yes: **Paul O. Adekola, PhD
